# Hybrid Films from Blends of Castor Oil and Polycaprolactone Waterborne Polyurethanes

**DOI:** 10.3390/polym14204303

**Published:** 2022-10-13

**Authors:** Gastón Pascual, Mirta I. Aranguren, Verónica Mucci

**Affiliations:** Instituto de Investigaciones en Ciencia y Tecnología de Materiales (INTEMA), Facultad de Ingeniería, Universidad Nacional de Mar del Plata-CONICET, Mar del Plata 7600, Argentina

**Keywords:** waterborne polyurethane, castor oil, tartaric acid, dimethylolpropionic acid, hybrid films, swelling, contact angle

## Abstract

Waterborne polyurethanes (WBPUs) with relatively high biobased content (up to 43.7%) were synthesized, aiming at their use as coatings for metals and woods. The study was performed on self-standing films obtained from anionic polyurethane water dispersions (PUDs). The initially targeted PUD was prepared from castor oil (CO), while tartaric acid (TA), a byproduct of wine production, was utilized as the internal anionic emulsifier. Although the films were cohesive and transparent, they were fragile, and thus blending the CO-TA PUD with other WBPUs was the chosen strategy to obtain films with improved handling characteristics. Two different WBPUs based on polycaprolactone diol (PCL), a biodegradable macrodiol, were prepared with dimethylolpropionic acid (DMPA) and tartaric acid (TA) as synthetic and biobased internal emulsifiers, respectively. The use of blends with PCL-TA and PCL-DMPA allowed for tailoring the moduli of the samples and also varying their transparency and haze. The characterization of the neat and hybrid films was performed by colorimetry, FTIR-ATR, XRD, DMA, TGA, solubility and swelling in toluene, and water contact angle. In general, the addition of PCL-based films increases haze; reduces the storage modulus, G’, which at room temperature can vary in the range of 100 to 350 MPa; and reduces thermal degradation at high temperatures. The results are related to the high gel content of the CO-TA film (93.5 wt.%), which contributes to the cohesion of the blend films and to the crystallization of the PCL segments in the samples. The highest crystallinity values corresponded to the neat PCL-based films (32.3% and 26.9%, for PCL-DMPA and PCL-TA, respectively). The strategy of mixing dispersions is simpler than preparing a new synthesis for each new requirement and opens possibilities for new alternatives in the future.

## 1. Introduction

Public concern about environmental issues has impulsed the rapid growth of the use of bioresources for the production of polymers and related materials. In the particular case of polyurethanes (PUs), the formulations have incorporated biopolyols to replace synthetic ones. Most frequently, these components have a vegetable origin, mostly based on plant oils [1,2,3,4]. These oils are a platform for the obtention of many different chemicals because of the wide variety of synthetic routes that can be applied to create new monomers, oligomers, and polymers. Diverse vegetable oils have been used, after chemical modification, as macrodiols or polyols, and several of them are now commercially available. Castor oil (CO) is the one most frequently used in the production of polyurethanes because it naturally contains hydroxyl groups that can readily react with isocyanates [5,6,7].

Following the path of reducing the environmental impact of the application of PUs, water-dispersible systems have been developed. While traditional PUs are vehiculized in organic solvents in applications such as coatings and adhesives, waterborne PUs (WBPUs), consist of polymer droplets that are dispersed in water. Consequently, the application and drying of these polymers result in water elimination during the films’ formation, reducing the problem related to the release of volatile organic components (VOC) into the environment, which also reduces the risks to human health during application [8,9,10].

Most frequently, ionic WBPUs are utilized, particularly anionic ones, because of their stability in suspension. In the synthesis of these polymers, an internal emulsifier is used consisting of a diol containing a carboxylic acid functionality and whose OH groups can react with the isocyanate to become integrated into the molecular structure of the polymer. The addition of a counterion component to the media creates the electrostatic layer needed to stabilize the suspension. One such anionic internal emulsifier of wide application is dimethylolpropionic acid (DMPA) [11]. An anionic and biobased diol, tartaric acid (TA), has also been proposed, having the characteristic of containing two acid groups, instead of one, as DMPA does. Although its careful incorporation results in the increase in biobased compounds in the formulation, it rigidizes the films obtained from these materials due to the partial reaction of one of the acid groups with isocyanate, which leads to a more tight-branched molecular structure [12]. The use of a biobased internal emulsifier is not common in the published literature, although Liu et al. [13] have discussed the effect of using a biobased synthesized emulsifier that, as in the case of the present manuscript, also reacts with a functionality greater than 2.0, becoming part of the polyurethane structure. This type of result leads to the competing effects of the emulsification and crosslinking induced by the same component. In previous publications, it was reported that the incorporation of biodegradable polycaprolactone (PCL) in the synthesis of a formulation prepared from CO and TA led to the production of films with lower modulus and less rigidity than the original CO-TA one [14].

It must also be considered that the size of the particles affects the velocity of the drying step after the application of the WBPU. Generally, large particles are used when rapid drying is needed, while small particles lead to longer drying times but allow the penetration of the PU in the substrate, which could be essential in some applications [15]. The structural molecular characteristics of the polyol, its functionality and molecular weight, the nature, characteristics, and concentration of the internal emulsifier, and the ratio of isocyanate to hydroxyl groups are factors that affect the properties of the dispersions and also of the final films [16]. 

As in other PU formulations, WBPU molecular structures can be considered segmented polyurethanes with hard and soft segments. The compatibility between the segments affects the mobility of the chains, which produces changes in the glass transition temperature of the material and can also affect the capability of the soft segments to arrange and form a crystalline phase.

The blending of different polymers to produce a dispersion better suited for a specific coating application has been considered, for example mixing WBPU with acrylic polymers [17]. A lower number of works analyze the mixing of different WBPUs; for example, Yen et al. investigated the effect of the addition of a PU prepared with polydimethylsiloxane to WBPUs prepared with different kinds of soft segments [18].

In the present work, the use of a biobased internal emulsifier was considered, thus CO and TA were reacted with isophorone diisocyanate (IPDI) to obtain a bio-WBPU with relatively high biological content. Mixing the obtained dispersion with two different PCL-based WBPUs was chosen as the strategy to vary the final properties of the casted films. One of the PCL-based WBPUs was prepared using TA as the internal emulsifier, while the second one was prepared with DMPA. The physical cohesion of the films and their partial crystallinity due to the presence of PCL were especially considered in the analysis of these mixtures, since not only the rigidity of the material can be varied by this simple preparation method but also the optical properties of the films and their thermal stability. Additionally, since the dispersions have potential application as coatings, the films were also characterized by their dissolution in toluene (gel fraction was determined), initial swelling in this organic solvent, and water surface affinity (water contact angle). Results obtained for these properties are presented in this work and the analysis is based on the relationships between the observed different properties and the structural differences among the neat WBPUs and their blends.

## 2. Experimental Materials

Isophoronediisocyanate (IPDI 98% purity. f = 2), triethylamine (TEA. 99% purity), dibutyltindilaurate (DBTDL 95% purity), dimethylolpropionic acid (DMPA 98% purity f = 2), dimethylformamide (DMF), and acetone were all purchased from Sigma-Aldrich Corp. and used without purification. L-(+)-Tartaric acid (TA 98% purity, f = 2) and toluene (P. A.) were purchased from BioPack S.R.L. Castor oil (CO supplied by Parafarm^®^, OH number = 177.21 mg of KOH/g. Iodine value = 82.36 g I_2_/100 g, f = 2.9) and polycaprolactone diol (PCL, provided by Sigma-Aldrich Corp. M_n_ = 2000) were used as the reagents contributing OH groups to the WBPU synthesis. The two carboxylic acids, castor oil and PCL, were dried in a vacuum before use. 

## 3. Preparation of the WBPU Suspensions and Films

Two of the WBPUs were synthesized following protocols previously developed by the group [12,19], using a 500 mL reactor, and were coded CO-TA and PCL-DMPA. Briefly, CO-TA was formulated using CO as the only polyol, tartaric acid as the internal anionic emulsifier, and IPDI as the isocyanate component. The molar ratios used were: NCO/OH = 1.6 y CO/TA = 0.8. The sample PCL-DMPA was synthesized from PCL macrodiol, DMPA as the emulsifier, and IPDI, using molar ratios of NCO/OH = 1.4 y PCL/DMPA = 1.6. In both cases, the counterion was TEA, added stoichiometrically to the concentration of anionic groups in the polymer. Details on the synthesis and characterization of the materials have already been discussed in a previous publication [12]. Additionally, a third polyurethane was synthesized for this work, using a procedure similar to the already-published protocol. Briefly, dry PCL and a solution of TA in DMF (30% wt./vol.), in an OH-equivalent molar ratio of PCL to TA = 1.6, were mixed in a glass five-necked flask of 250 ml equipped with a condenser, a mechanical stirrer, and a flow inlet to allow the entrance of nitrogen. IDPI in a ratio of NCO/OH = 1.4 and DBTDL (1 wt.% with respect to the total reaction mass) were then added, and the reaction proceeded for 4 h at 85 °C. At this stage of the polymerization reaction, the viscosity increased; therefore, dry acetone was added to maintain a constant value. After 4 hours of reaction, the temperature was reduced to 60 °C, and TEA was added in an equimolar ratio with respect to TA to form the salt that allowed subsequent dispersion in water. After 30 min, the temperature was reduced to room temperature, and 100 mL of distilled water was slowly added under vigorous stirring (800 rpm). At the end of the addition, the agitation was reduced to 200 rpm and was maintained overnight. The remaining acetone in the obtained suspension was removed by rotary vacuum evaporation at 30 °C. This WBPU was coded PCL-TA.

The hybrid suspensions were prepared by the direct mixing of the WBPUs in the desired proportion. Each suspension obtained was then stirred for 30 min in an ultrasonic bath. As shown, the formulations were named according to the base polyol and the emulsifier used. Numbers in parentheses correspond to the weight percentage used in the formulation of the blends. Thus, CO-TA(60)/PCL-TA(40) indicates that the sample was prepared by mixing CO-TA and PCL-TA in a weight proportion of 60 to 40 (dry solids basis).

For the film preparations, the suspensions (WBPUs neat and blends) were cast in glass Petri dishes coated with nonstick adhesive paper. The suspension volumes used resulted in films with thicknesses of 200 to 340 microns (Appendix A). The samples were oven dried at 35 °C overnight. 

## 4. Characterization

### 4.1. DLS (Dynamic Light Scattering)

The average particle size of the neat WBPU dispersions was measured using a Malvern Zetasizer Nano S-90 with a laser beam of 632 nm (Malvern Instruments Co. Ltd., Worcestershire, United Kingdom). The measurement was carried out at room temperature, and the samples were previously diluted in distilled water; three measurements were taken for each WBPU synthesized.

### 4.2. Colorimetry

The opacity of the films as well as the color coordinates were measured using a Lovi Bond Colorimeter RT500 (Amesbury, United Kingdom) with an 8 mm diameter measuring area. Color differences between the films were also quantified. The measurements are based on the readings of the values *L**, *a**, and *b**, from which the white index (WI) is calculated using the equation:(1)$$\mathrm{WI}=100-\sqrt{{\left(100-L^{*}\right)}^{2\;}+{a^{*}}^{2}+{b^{*}}^{2}}$$
where the value of *L** indicates brightness; the larger the value of *L**, the brighter the sample. The value of *a** corresponds to the red–green axis, with positive values indicating a larger contribution of the red color and negative values indicating a larger green contribution. The value of *b** corresponds to the yellow–blue axis, with positive values indicating a larger contribution of the yellow color and negative values indicating a larger blue contribution.

### 4.3. Fourier Transformed Infrared Spectroscopy (FTIR)

The spectra of all samples were obtained at room temperature using a Thermo Scientific Nicolet 6700 spectrometer using an attenuated total reflectance accessory (ATR) with a diamond crystal. The spectra were collected as the average of 64 scans, with a resolution of 4 cm^−^^1^ in the wavelength range of 400–4000 cm^−^^1^. 

### 4.4. X-ray Diffraction (XRD)

The X-ray spectra were obtained using an X PANalytical X’ Pert PRO X-ray diffractometer (Cu Kα radiation with wavelength: 1.54187 Å). The range covered was from 5° to 60° and the scanning speed was 0.016° s^−1^. All measurements were carried out one week after obtaining the films. Publicly licensed software [20] was used in the deconvolution of the peaks, which allowed us to calculate the percentage of crystallinity of the samples as 100 × (total area-area of the amorphous halo)/total area).

### 4.5. Thermogravimetric Analysis (TGA)

The thermal degradation of the films was investigated using the thermogravimetric method. The measurements were performed on 8–10 mg samples using a TGA-50 Shimadzu (Japan) under an N_2_ atmosphere, from room temperature to 700 °C at 10 °C/min.

### 4.6. Dynamic Mechanical Analysis (DMA)

The tests were performed using an Anton Paar Physica MCR 301 rheometer, in torsion mode, one week after the films were prepared and stored in a dessicator. The parameters used for the test were: a fixed frequency of 1 Hz and a small oscillation amplitude of 0.05%. Three repetitions were made per sample, the size of which was 10 × 2 × 30 mm. The damping factor (tan δ) and storage shear modulus (G’) were measured as a function of temperature, at a heating rate of 3 °C/min.

### 4.7. Gel Content and Swelling in Toluene

Pieces of the films of approximately 1 cm^2^ were packed inside Watman No. 4 filter paper bags and then immersed in glass containers filled with toluene. The samples were left for 9 days, with the daily removal and replacement of the solvent. After that time, they were extracted and the weight of the bags was recorded until reaching a constant weight. Duplicate measurements were performed for each sample. The residual material contained in each bag corresponded to the gel fraction of the sample.

For the study of swelling, neat and hybrid WBPU films were immersed in toluene for 50 h at room temperature. At different time intervals, the samples were removed, blotted dry with filter paper, and then weighed. At least three replicate measurements were performed per sample. The solvent absorption was calculated as the percentage of weight increase with respect to the initial dry weight.

### 4.8. Contact Angle

Measurements of the static contact angle were made using bidistilled water, using a goniometer OCA 15LHT Plus photo-microscope, Dataphysics, with a high-resolution camera, and Microsoft Photo Editor software. A drop of water was deposited on the surface of the film and the angle formed between the surface of the WBPU film and the tangent line to the drop of liquid was measured using the software at 10 readings per second. Measurements were performed in triplicate. 

### 4.9. Statistical Analysis

Statistical analyses were performed using free MedCalc statistical software (accessed on 11 August 2022 https://www.medcalc.org/calc/comparison_of_means.php) using the *p*-value to indicate significance level (*p* < 0.05 indicating that two means are significantly different).

## 5. Results and Discussion

### 5.1. Suspensions Characterization

All the samples were stable over a period of several months. A photographic register of test tubes containing the samples was taken during the first 45 days. As in the case of the neat polyurethanes, no sedimentation and no macrophase separation between the WBPUs was observed in any of the tubes (Appendix A), meaning that the blending of the WBPUs did not have a negative effect on the stability of the suspensions. The observation is in agreement with reported results on WBPU suspensions based on vegetable oils that presented good stability with particles in a wide range of sizes (~80 nm to 825 nm) [21].

The sizes of the particles present in the neat polyurethane suspensions were determined by dynamic light scattering (DLS). CO-TA showed the smallest particles at 38.42 ± 0.97 nm, while the PCL-TA and PCL-DMPA showed particles of 278.03 ± 6.49 nm and 179.4 ± 3.66 nm, respectively. It has been shown that the size of the particles is dependent on the polyol molecular weight and structure, the ratio of NCO/OH, as well as the preparation conditions (particularly, the water and stirring step). In the present case, the water addition was performed in similar conditions for all the samples, thus the differences in particle size should be the result of the variation of the other parameters. 

Regarding the effect of the molecular weight of the polyol, Bhattari et al. [16] prepared WBPU from polytetramethylene glycols (PTMGs) of different molecular weights and reported that the average size of the particles increased from 860 nm to 1355 nm for PTMGs of molecular weights of 1000 and 2000 Da, respectively. Thus, other conditions being equal, the increase in the length of the chain diol led to the increase in the average particle size of the WBPU. Further increases in the molecular weight resulted in a broad particle size distribution. In the present case, the molecular weight of CO is about half that of the PCL, but additionally, the latter is a linear molecule with reactive end groups and CO is a triglyceride (it can be envisioned as a three-arm star molecule) with reactive groups in the middle of the arms. Since the ratio of OH coming from the CO to those coming from the TA is 0.8, it also means that the concentration of emulsifier is relatively high, which is necessary to avoid the gelation of the PU during the first step of the synthesis. These factors appear to be the most important in determining the relatively small size of the CO-TA particles. Similar results were reported by Liu et al. [13], who also worked with a synthesized bioemulsifier capable of partial reaction with the isocyanate. In their case, the authors also reported that the effect of the emulsifier was more important than the crosslinking effect, resulting in small particles.

On the other hand, the different sizes of the particles of PCL-TA and PCL-DMPA can be explained by the fact that TA can react with the isocyanate via the two hydroxyl groups, but also partially via one of its carboxylic acid groups. Thus, the PCL-TA polymer presents a less flexible structure, which results in larger particle sizes. The effect of crosslinking on the size of the particles has already been discussed for other systems in the literature [12,15].

### 5.2. Films Characterization

Figure 1 shows the digital images of the films showing that the samples were transparent or translucent with a yellow hue when they were prepared from CO and more opaque if they contained a PCL-based WBPU [22]. Besides the transparency observed when the films are placed in contact with the supporting surface (Figure 1a–e), the images also illustrate the light dispersion (haze) of the films (Figure 1f–j), a characteristic that may prove valuable to reduce glare [23].

The color differences between the samples were also quantified using colorimetry. The numerical results and the corresponding statistical analysis are reported in the Appendix A. 

The measurements were performed one week after the obtention of the films. The colorimetric results confirmed the qualitative observations in that the neat CO-TA film has high transparency (low opacity) and a yellowish color, which are characteristics of vegetable oil-based coatings [24]. Moreover, the opacity was higher for the PCL-based films, showing a milky appearance. Regarding the blend films, those containing PCL presented a higher opacity than CO-TA, due to the partial crystallization of the macrodiol segments [25]. Although the dispersion of the results obtained for the PCL-TA sample was large, general trends can be observed. Overall, the hybrid films that contain a higher percentage of PCL are more opaque, particularly those prepared using DMPA (Figure 2). This is attributed to the lower crosslinking of the WBPU prepared with this acid, which enables the greater mobility of the PCL segments to arrange into a crystalline phase, as will be further discussed. 

### 5.3. Infrared Spectroscopy

The FTIR spectra of the films (Figure 3) show the characteristic peaks of this type of polymer: at 3370 cm^−1^, the stretching vibration band –N–H of the urethane groups; at 1728 cm^−1^, the stretching vibration of the carbonyl in the amide-I region; and at around 3000 cm^−1^, the characteristic vibrations of C-H stretching [26]. In the case of the neat WBPUs, the band which corresponds to the N-H stretching is more intense for the CO-TA, which was expected because of the higher functionality of the triglyceride (f = 2.9) that leads to the formation of more urethane bonds. This is confirmed by the increased intensity of the band at 1540 cm^−1^, which corresponds to the amide-II absorption related to the stretching of C-N, C-C bonds, and -NH in-plane bending [27]. In the spectra of the WBPUs containing PCL, the band corresponding to the carbonyl group is narrow and intense, in agreement with previous observations for this type of polymer [19,28]. In the case of the formulated mixtures, it is observed that as the percentage of PCL-based WBPU decreases, the peak corresponding to the carbonyl decreases, which is understandable since PCL is a polyester, and as explained above, it presents a very intense peak of the carbonyl group. Additionally, a difference is observed in the formulations that use TA as an emulsifier due to the peak that appears at 1650 cm^−1^, which corresponds to the absorption of the carboxyl group of the tartaric acid [12].

### 5.4. X-ray Diffraction

Figure 4a,b show the X-ray diffraction spectra of the samples containing different concentrations of the WBPUs. The films made from CO-TA are amorphous, as was expected, considering the structures of the polyol and emulsifier. On the other hand, all samples containing PCL diol showed some crystallinity, regardless of the emulsifier used. This observation is in agreement with the higher opacity of the samples containing PCL segments, as has already been reported in a previous section.

Figure 4c shows the results of the percentage of crystallinity of the samples vs. the content of the PCL-based WBPUs. As was expected, the crystallinity increased with the amount of the PCL-based WBPU contained in the formulations for the two sets of blends and covers a rather large range of values, an observation reported previously on PCL-based WBPUs [29]. The form of the curves (drawn only as guiding lines) shows that the presence of the more crosslinked CO-TA sample interferes with the crystallization of the PCL segments, and the effect is more important when the concentration of CO-TA is high. This observation is in agreement with the results of Yang et al. [29], who worked with WBPU based on PCL and reported that the higher flexibility of the molecular structure of the PU was related to the easier crystallization of the soft PCL segments. 

There is also a difference between the samples, which depends on the internal emulsifier utilized in the synthesis. The difunctional, monoacid DMPA allows the formation of linear chains in PCL-DMPA, while the use of TA as an internal emulsifier in PCL-TA can lead to the formation of linear as well as branched chains. The fact that TA has two acid groups has been already shown to result in the reaction of some of the acid groups with the isocyanate [12]. Consequently, some of the chains in the structure of this WBPU are not linear, resulting in the lower mobility of the molecular chains, and thus, the lower crystallinity. To summarize, crystallinity depends on at least two factors: the type of emulsifier used and the concentration of the PCL-containing WBPU in the sample. As the content of PCL segments increases, the percentage of crystallinity in the sample increases because those are the only crystallizable segments in the WBPU structure. Additionally, as the content of CO-TA increases, the mobility of the chains decreases, leading to lower crystallinity. The difference between the two emulsifiers becomes more important at high contents of PCL-based WBPU.

On the other hand, the structure of the crystallites is essentially not affected by the presence of CO-TA, since the position of the main crystallization peak of the PCL does not vary in position and varies only slightly in width (width at the medium height of the peak). Considering all the samples, the position of the main crystalline peak is 21.53 ± 0.14° and its width is 0.397 ± 0.061°. The complete set of results is included in the (Appendix A).

### 5.5. Thermal Gravimetric Analysis 

Although the comparison of the three neat WBPU shows that the degradation of CO-TA begins at a lower temperature, the entire event occurs along a wider temperature interval. Other authors have also commented on the beginning of decomposition at lower temperatures in the case of short polyols due to the closer urethane linkages that present lower thermal stability [30]. The particular thermal degradation profile of the CO-TA type of polymer has also been discussed in a previous publication [12]. Briefly, a study of the thermal decomposition of the salts formed between DMPA or TA with the TEA counterion showed that it occurred at a markedly lower temperature in the case of TA. Thus, the initial decomposition (below ~250 °C) was assigned to this decomposition step, Figure 5. In the present case, after this initial step, a second occurs between 250 and 400 °C, which corresponds to the decomposition of urethane bonds, followed immediately by the decomposition of the biopolyol. Finally, the last step takes place between 420 and 510 °C and involves the decomposition of the materials resulting from the previous degradation steps [12,29,31]. The values of the peaks obtained from the maxima in the dTG peaks were 213 °C, 344 °C, and 460 °C, respectively (Appendix A).

As already mentioned, the thermal degradation of the PCL-based WBPU begins at a higher temperature than that of CO-TA, in agreement with the observations of other authors [32], who have reported that as the polyol functionality increases or the length of the soft segments decreases, the concentration of urethane groups consequently increases and the thermal stability of these polymers decreases. Thus, the low stability of urethane bonds is more important than the effect of a more developed crosslinked structure [13,29]. 

The main step of the thermal degradation of PCL-TA and PCL-DMPA occurs in a narrow temperature range of around 310–350 °C. The event takes place at a lower temperature for the PCL-TA sample as compared to the PCL-DMPA, which is related to the different chemical nature of the emulsifier that leads to the already-discussed lower crystallinity of the PCL-TA.

The TG curves obtained for the hybrid films fall in between those of the neat WBPUs and closer to that of the CO-TA, which is the major component in the studied mixtures. Table 1 summarizes the overall behavior, reporting the temperatures registered when the weight losses are 10, 50, and 95%. The final char is negligible for all the samples considered.

### 5.6. Dynamic Mechanical Analysis

#### 5.6.1. Neat WBPUs

Figure 6a shows the storage moduli and tan δ curves of the films made from the neat WBPUs used in the study. Measurements were performed one week after preparation, to ensure that the slow crystallization of the PCL segments had already taken place. In this regard, Yang et al. [29] have also reported that the crystallization of PCL segments was not immediate when attached to the WBPU structure. Instead, the first cooling and second heating in the DSC showed no crystalline phase, although, during the film formation (casting/drying), a crystalline phase had developed, pointing to the fact that developing a crystalline PCL phase in the films is a relatively slow process [19].

At low temperatures, the storage moduli of the neat WBPU films containing PCL show higher values than CO-TA samples, due to the additional effect of the crystalline phase of PCL and the absence of the plasticizing alkyd chain ends of the castor oil, which are present in the corresponding film. When comparing the behavior of the two neat films based on PCL, differences also appear due to the selected emulsifier. Although in both cases the PCL segments can crystallize, the use of TA instead of DMPA introduces some degree of crosslinking/branching in the structure without much affecting the linear PCL segments. The overall effect is a higher storage modulus at low temperatures.

As the temperature increases, the PCL effect is less important; the curves follow a similar decay rate until reaching a temperature close to 40 °C, at which the melting of the crystalline phase (PCL segments) occurs, and since the chains are linear (PCL-DMPA) or have a very low number of crosslinking points (PCL-TA), the material becomes liquid-like and the modulus drops catastrophically. It must be noticed that although high molecular weight PCL has a melting temperature of around 60 °C, the PCL used in this study is relatively short (MW = 2000 Da) and the chain segments are part of the reacted structure, so the formation of perfect crystals is restricted, which would explain the lower melting temperature observed for the films. This result is in agreement with previous results, in which the melting event occurred at a lower temperature as compared with that of neat long molecular weight PCL [19,33]. On the other hand, the CO-TA film is crosslinked and amorphous and since no melting event takes place, the modulus maintains relatively high values well above room temperature [12].

The analysis of the tan δ curves gives similar information about the samples (Figure 6a). When the temperature reaches the value of the melting point of the PCL segments, the curves of the neat PCL-based films show a steep growth, indicating that the materials become liquid. The differences in the curves of these two materials at low temperatures are related to the different internal emulsifiers used. As analyzed in the XRD section, PCL-TA films are less crystalline than the PCL-DMPA ones, which results in different relaxation events at low temperatures. As for the CO-TA sample, the comparison shows that this sample always has a larger solid-like contribution than the other two WBPU, since tan δ is lower practically in the whole temperature range.

#### 5.6.2. Films Obtained from the WBPU Blends

*CO-TA/PCL-TA:*Figure 6b shows the storage moduli of the films prepared from CO-TA/PCL-TA at different weight ratios. At temperatures above the melting point of PCL, the behavior of the samples is similar to that of the CO-TA, although showing lower values of G’ because of the presence of the molten phase of the PCL-TA. It illustrates the benefits of the presence of CO-TA that impedes the catastrophic drop in the moduli of the films. On the other hand, at low temperatures, all the hybrid films show a higher modulus than the neat CO-TA due to the presence of the PCL crystalline phase. There is a large difference between the samples containing PCL-TA at 40 wt.% and 30 wt.%; while the first one closely follows the behavior of the neat PCL-TA (at low temperatures), the last one presents a much lower modulus, close to that of neat CO-TA. Thus, depending on the concentration chosen, the overall behavior is controlled by the presence of one of the two WBPUs used in the mixture. Besides the melting event of the PCL phase, the curves show that there is a minor thermal transition at around −50 °C and another at about 70 °C. The first one is related to the PCL-rich phase (neat PCL has a glass transition temperature of around −60 °C, but appears shifted by constrictions to chain mobility in the WBPU) [19,34] and the second transition corresponds to a relaxation in the CO-TA network. As is already known, the relaxations of the polymers observed in DMA studies are related to the glass transition temperatures, T_g_, of the materials and commonly used in its measurement, even if it is also affected by the conditions of the dynamic mechanical measurement and not only by the temperature scan [35].

Figure 6c shows in more detail the tan δ curves at low temperatures. It can be seen that as the concentration of PCL-TA is reduced, the larger concentration of CO-TA shifts the temperature of the relaxation of the PCL-rich phase towards higher temperatures. The absolute value of the maxima in the curves is also reduced and the width of the transition is increased, which indicates that the CO-TA is restricting the mobility of the chains and also that fewer separated microphases are obtained, which leads to the broadening of the transition [36]. In this regard, it has been reported that when the crosslinking density increases, the mobility of the chains is more restricted, leading to higher glass transition temperatures. Furthermore, a broadening and a decrease in the intensity of the tan δ peak were observed, which were also related to the restricted chain mobility [35,36]. Other authors observed that when an emulsifier with a branched structure (produced by the partial reaction of carboxyl groups in the molecule) was used, it led to the reduction in the packing capability of the macrodiol segments and to a polyurethane with more mixed phases [13]. In the present study, these effects are seen when the percentage of CO-TA is increased.

*CO-TA/PCL-DMPA:* The storage moduli of the CO-TA/PCL-DMPA films are shown in Figure 6d. The behavior is similar to that discussed for the CO-TA/PCL-TA films, but the moduli are lower, in agreement with the lower storage modulus of PCL-DMPA with respect to PCL-TA. As in the previous case, the use of CO-TA allows the maintenance of the structure of the films at temperatures above 50 °C. The values of tan δ are higher than those measured for the films CO-TA/PCL-TA, indicating that the chains have higher mobility when the emulsifier is DMPA (Figure 6e). The maxima of the curves in the low-temperature region are not very clear, with the exception of the samples containing 20 and 30 wt.% of PCL-DMPA, appearing at a temperature close to that obtained for CO-TA(70)/PCL-TA(30). 

### 5.7. Solubility and Swelling in Toluene

#### 5.7.1. Solubility Test and Gel Fraction

Figure 7a shows the gel content in the films as the average value, with the error bars representing the minimum and maximum values measured for each sample.

The theoretical expectations were that the CO-TA films would be completely insoluble and the PCL-containing films completely soluble in toluene. As shown in Figure 7a, the experimental results were close to the expected values for the neat WBPU films. Clearly, the existence of a soluble fraction results from the existence of an uncrosslinked structure, as has been reported by other authors [17,36].

With respect to the mixtures, it was initially expected that the fractions of crosslinked CO-TA were insoluble and the fractions corresponding to the PCL-based WBPUs were almost completely soluble [17]. Thus, in the films produced from the blend of the suspensions, the gel percentage would be close to that of the percentage of the CO-TA in the sample. Figure 7a shows that for the lowest content of CO-TA (60 wt.%), the percentage of gel measured was very close to the expected value: ~61% and 59% for the CO-TA(60)/PCL-TA(40) and CO-TA(60)/PCL-DMPA(40) samples, respectively. As the content of CO-TA increases, the overall behavior suggests that CO-TA interacts with or encloses the minor WBPU component, restricting the extraction, so that the gel% becomes higher than the expected value. For example, at 80 wt.% of CO-TA, the values become ~86% and 88% for the CO-TA(80)/PCL-TA(20) and CO-TA(80)/PCL-DMPA(20) samples, respectively.

Figure 7b shows that a single correlation can be considered between the % of crystallization and the gel%, independent of the PCL-based WBPU used. Obviously, the higher the gel fraction (corresponding to a higher CO-TA concentration), the lower the crystallinity of the sample. 

#### 5.7.2. Swelling in Toluene

Figure 7c shows the absorption of toluene, presented as a percentage of increased weight with respect to the initial value. The plot shows the relation with the percentage of gel in the sample, and the curves were drawn only as a guide to the eye.

Regarding the neat WBPUs, the two samples containing PCL dissolve in toluene (as already discussed), so swelling was not measured for those two samples. As for the CO-TA film with a gel fraction close to 100%, the percentage of absorbed toluene was above 40%, and the value grew as the time of immersion increased, which is the expected and usual behavior.

The reduction in the “nominal” toluene absorption of the hybrid films is the result of the simultaneous absorption of toluene and the dissolution of the PCL-containing fraction, a logical result already reported in the literature on polyurethanes [37]. For that reason, the results at a relatively short time (500 min) are presented, together with the results obtained at a longer time (2900 min), illustrating the effect of the dissolution of the sample. This is the reason for the measured values (% weight increase) decreasing after an initial period of growth. It is interesting to notice that the samples prepared with PCL-TA absorb less toluene than PCL-DMPA at any given time. This is related to the lower mobility of the branched chains of PCL-TA and its higher compatibility with CO-TA.

### 5.8. Water Contact Angle

Water contact angle (WCA) measurements were carried out and the results are shown in Table 2 (representative images are shown in Appendix A). The relatively low hydrophilicity of the CO-TA films is in agreement with results reported by Liang et al. 2018, who worked with WBPUs synthesized from vegetable oils and found that the polyols with higher functionality presented higher water contact angles. They also discussed that in their particular case, although the larger amount of OH groups led to the increased concentration of the internal emulsifier, which should reduce the contact angle, the effect of the increased crosslinking density compensates for the previous effect and the overall result was a higher water contact angle. In the present case, the same general result was observed for the CO-TA film [32]. As seen in the table, all formulated mixtures are more hydrophilic (lower contact angle) than CO-TA, especially the mixtures formulated with PCL-TA [36,38,39]. This is related to the higher hydrophilic nature of the surface of PCL-TA, which could be explained by the presence of the two carboxylate groups of the diacid in the original TA molecule, a result that has been also previously reported [38].

## 6. Future Work

The strategy of blending appears as a versatile method for tailoring the properties of the final films. Different from changes obtained through variations in the synthesis formulation that require a new formulation for each variation proposed, the blending of already-prepared WBPUs is a more simple preparation. However, the stability of the mixed dispersion should be initially verified, and in the present case, the mixing of the original suspensions did not pose any problem. An interesting point to be considered in the future is the controlled mixing of dispersions that have different and controlled particle sizes. This could be used to study the haze produced by the dispersion of the large particles through the film, which would not require the addition of other non-PU components to produce matte coatings of reduced glare. Additionally, the resulting blend could show lower viscosity than the original suspensions because of the already-known fact that differently sized particles can pack more closely together than monodisperse ones [40]. This last observation could also lead to the formulation of blend dispersions of relatively low viscosity but with high solid content (a strategy that has already been used in different suspension systems) [41], which may be useful to obtain formulations with shorter drying times.

## 7. Conclusions

The preparation of a WBPU with a high content of the biobased material CO-TA led to films that are hard but have relatively low flexibility. The present study showed that it is possible to use the simple blending of different WBPUs to modify the properties of the original neat material. This strategy is much simpler than tailoring the formulation, which would require a new synthesis for each change in the percentage of the flexible macrodiol, PCL. The blending technique showed to be effective, did not negatively affect the stability of the suspensions, and allowed us to control the final properties of the films by varying the proportion of the already-prepared neat suspensions.

The study showed that the hybrid films were transparent or translucent with a matte surface. The fact that they show haze is proposed as an advantage for obtaining antiglare coatings. The hybrid films showed improved cohesion and resistance to toluene with respect to the PCL-based WBPUs. The storage moduli at room temperature were smaller in the hybrid films compared to CO-TA, which is related to the lower T_g_ of the films containing PCL, thus also allowing variation in this property. 

The content of gel and crystallinity were important factors that affected the final properties of the films. Furthermore, the nature of the emulsifier used in the synthesis of the PCL-based WBPUs had also an effect on the crystallinity of the samples, and according to the thermal relaxations observed by DMA, compatibility was higher in the case of the CO-TA/PCL-TA films (broader relaxations). The surface hygroscopicity of the neat CO-TA was increased by mixing it with any of the two PCL-based WBPUs, particularly with PCL-TA. 

To summarize, a simple strategy of blending WBPUs was shown to be applicable to modify the haze of the films, the storage modulus, the resistance to toluene, and the surface hygroscopicity by simply changing the ratio of the WBPUs in the blends. The mixing of different WBPUs would offer an easy route to the tailoring of specific properties if the stability of the mixed dispersion is confirmed.

## Figures and Tables

**Figure 1 polymers-14-04303-f001:**
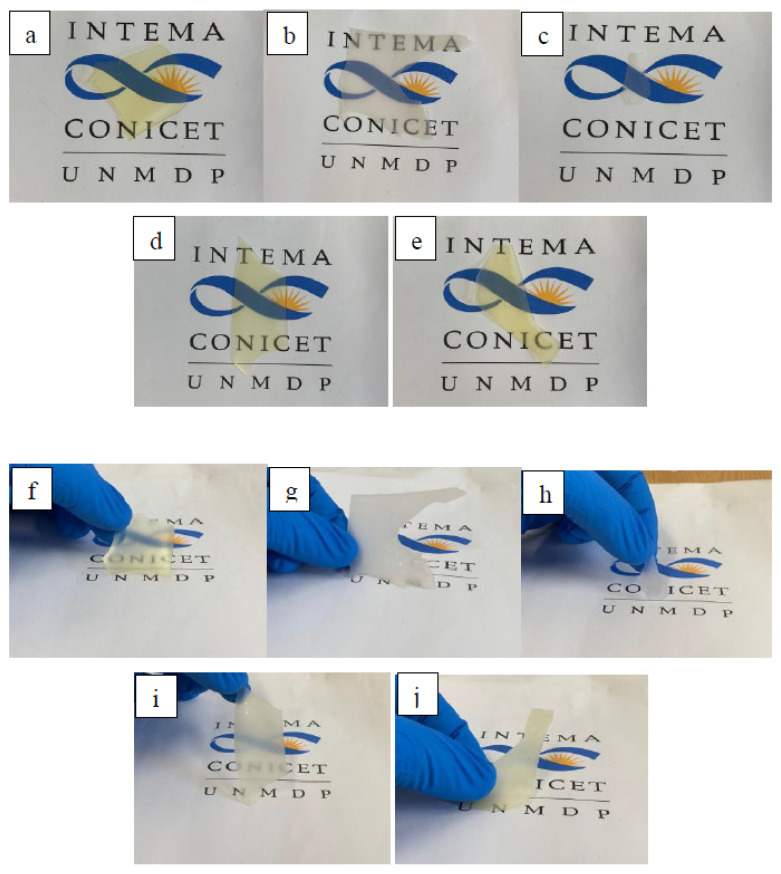
WBPU films: (**a**,**f**) CO-TA. (**b**,**g**) PCL-DMPA. (**c**,**h**) PCL-TA. (**d**,**i**) CO-TA(80)/PCL-TA(20). (**e**,**j**) CO-TA(80)/PCL-DMPA(20).

**Figure 2 polymers-14-04303-f002:**
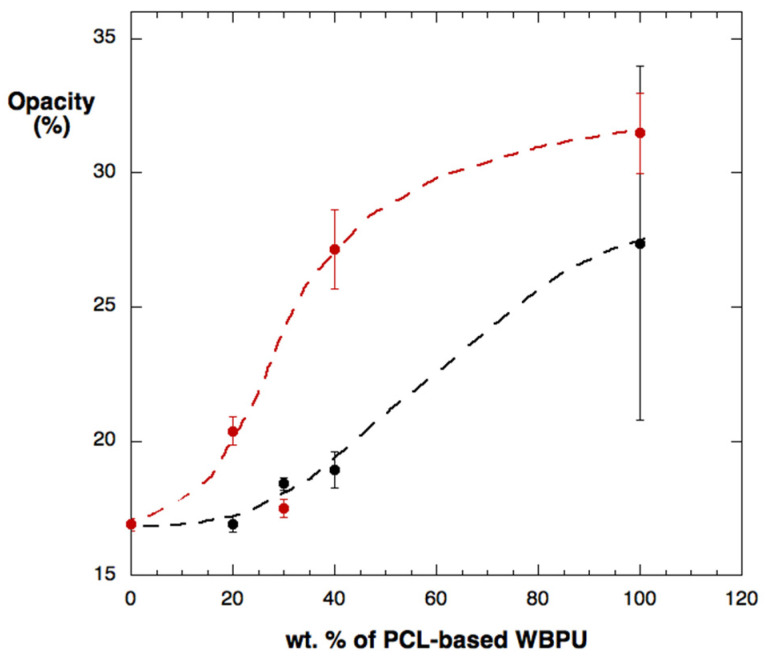
Opacity of the two sets of hybrid films: CO-TA/PCL-TA (black circles) and CO-TA/PCL-DMPA (red circles). Lines are drawn as guides to the eye.

**Figure 3 polymers-14-04303-f003:**
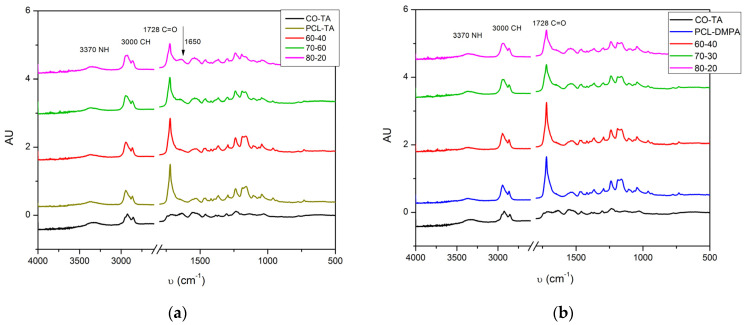
FTIR spectra of the obtained WBPUs and the formulated mixtures: CO-TA/PCL-TA samples (**a**) and CO-TA/PCL-DMPA samples (**b**).

**Figure 4 polymers-14-04303-f004:**
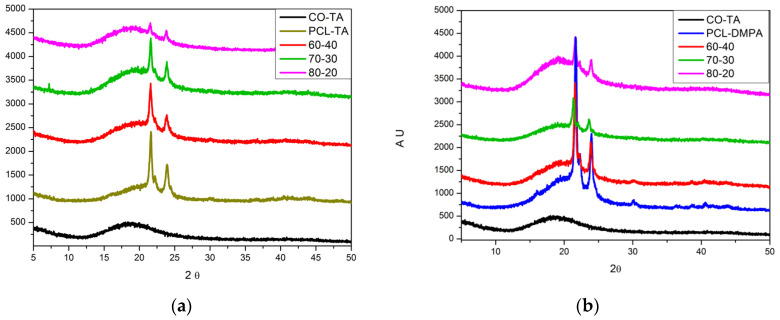
Diffraction patterns of samples containing different contents of PCL-TA (**a**) and PCL-DMPA (**b**). Effect of the content of PCL-based WBPUs on the crystallinity of the samples (**c**). Lines are drawn as guides to the eye.

**Figure 5 polymers-14-04303-f005:**
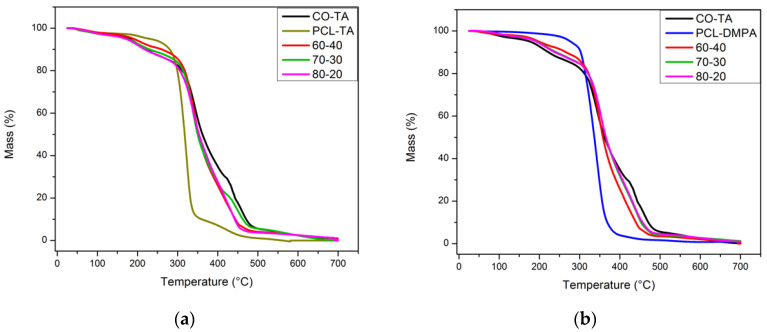
Thermogravimetric results for the hybrid films containing PCL-TA (**a**) and PCL-DMPA (**b**).

**Figure 6 polymers-14-04303-f006:**
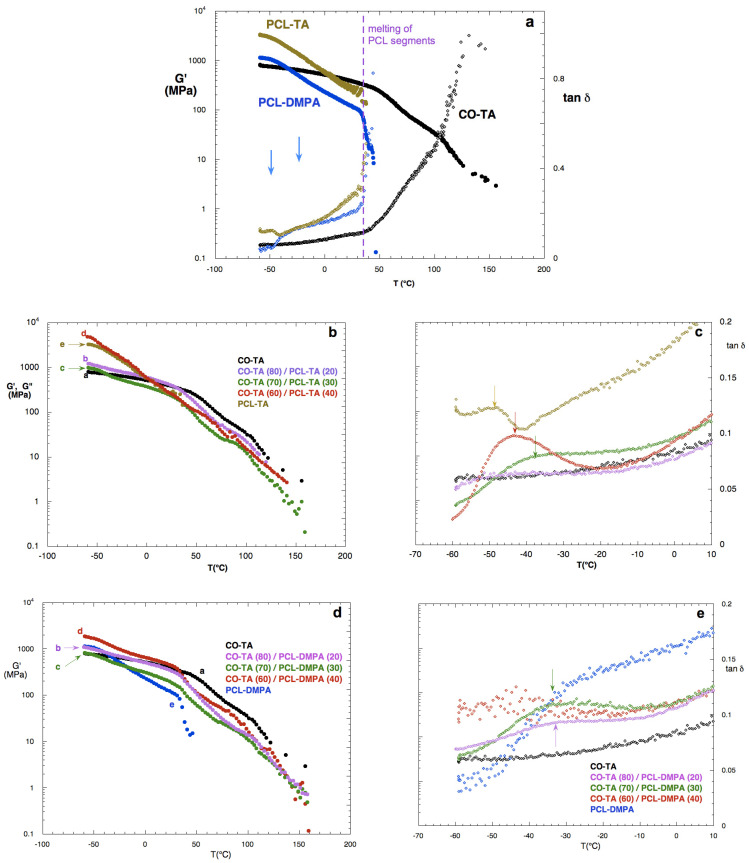
Storage moduli and tan δ curves for neat WBPUs (**a**), CO-TA/PCL-TA films (**b**,**c**), and CO-TA/PCL-DMPA films (**d**,**e**).

**Figure 7 polymers-14-04303-f007:**
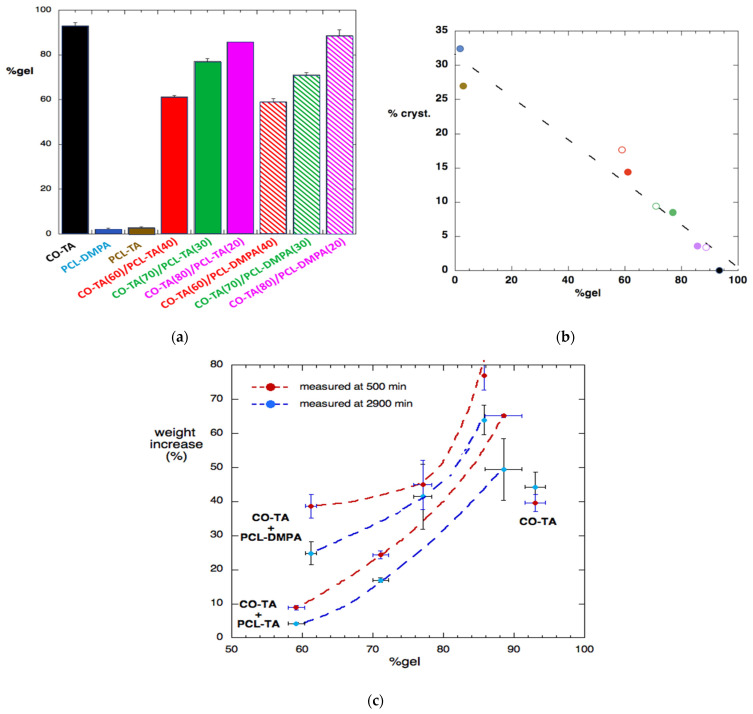
Gel content (%) of the films, extracted with toluene (**a**). Variation of the sample crystallinity with the gel% (**b**). The colors identify the same samples as in (**a**), with filled symbols used for the CO-TA/PCL-TA and open symbols for the CO-TA/PCL-DMPA. Toluene absorption by immersion in the solvent, calculated as % of weight increase (**c**) vs. the gel content of the samples. Lines are drawn merely as guides.

**Table 1 polymers-14-04303-t001:** Temperature (°C) at which the weight loss is 10%, 50%, and 95%.

	T_10_, °C	T_50_, °C	T_95_, °C
**CO-TA**	222	362	471
**PCL-TA**	281	316	459
**PCL-DMPA**	304	336	369
**CO-TA/PCL-TA (60–40)**	265	355	444
**CO-TA/PCL-TA (70–30)**	231	351	463
**CO-TA/PCL-TA (80–20)**	218	352	444
**CO-TA/PCL-DMPA (60–40)**	266	358	440
**CO-TA/PCL-DMPA (60–40)**	243	366	453
**CO-TA/PCL-DMPA (60–40)**	238	366	457

**Table 2 polymers-14-04303-t002:** Water contact angle measured on the films’ surfaces. Different letters correspond to samples that show statistically significant differences.

Sample	Water Contact Angle
CO-TA	73.14 ± 0.19 ^a^
PCL-DMPA	63.39 ± 0.07 ^b^
PCL-TA	50.19 ± 0.13 ^c^
CO-TA/PCL-TA 60–40	40.1 ± 0.49 ^d^
CO-TA/PCL-TA 70–30	54.20 ± 0.29 ^e^
CO-TA/PCL-TA 80–20	54.98 ± 0.28 ^e^
CO-TA/PCL-DMPA 60–40	63.69 ± 0.43 ^b^
CO-TA/PCL-DMPA 70–30	60.8 ± 0.34 ^f^
CO-TA/PCL-DMPA 80–20	61.28 ± 0.12 ^f^

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
