# Peer review of "Hybrid Films from Blends of Castor Oil and Polycaprolactone Waterborne Polyurethanes"

_polymers, 2022, doi:10.3390/polym14204303_

Round 1

Reviewer 1 Report

Overall a good presentation of blending materials to produce different properties of WBPU films. The experimental work and analysis methods were well described. The reasons for different properties obtain were well reasoned. Some minor spelling/grammar issues need to be addressed...

Ln 40-41: consist on (big gap which needs to be removed) polymer...change to exist as polymer

Ln 44: the risks to the human health... remove the the next to human

Ln 46: syntesis should be synthesis

Ln 47: consisting in a diol.... change in to of

Ln 229: independently should be independent

Ln 237: As it was expected....omit the it

Ln 251-252: as it could be expected.....remove it could be

Author Response

Reviewer #1

Overall a good presentation of blending materials to produce different properties of WBPU films. The experimental work and analysis methods were well described. The reasons for different properties obtain were well reasoned. Some minor spelling/grammar issues need to be addressed...

Authors:

Thank you for your positive comments. All the changes in spelling/grammar were addressed (the number of the lines were they appear are different in the revised version, where they are shown in red font).  The only different replacement from what was suggested was that the word "independently" was replaced by "regardless", still maintaining the meaning of the sentence (now appearing in line...).

Reviewer 2 Report

The paper Hybrid films from blends of castor oil and polycaprolactone waterborne polyurethanes try to find and describe the solution to environmentally friendly production of waterborne polyurethans-based coating films. The topic of the study is interesting and the experiments were well designed and described. However, some issues require improvements before publication in the Polymers journal. For this purpose see the comments below.

Please verify the whole manuscript for editorial issues. Pay attention to the font style, as it requires unification in many places in the manuscript, including the abstract.

It is not clear why you test CO-TA, PCL-DMPA, and PCL-TA, but not CO-DMPA. Please explain it.

Please improve the quality of fig. 3 and 4 (diffraction patterns)

The results and discussion section lacks discussion of the results. It is too scarce. Please improve it.

Please carefully verify the whole manuscript for editorial and language issues, paying attention to the literature section.

Author Response

Reviewer #2

The paper Hybrid films from blends of castor oil and polycaprolactone waterborne polyurethanes try to find and describe the solution to environmentally friendly production of waterborne polyurethans-based coating films. The topic of the study is interesting and the experiments were well designed and described. However, some issues require improvements before publication in the Polymers journal. For this purpose see the comments below.

 Please verify the whole manuscript for editorial issues. Pay attention to the font style, as it requires unification in many places in the manuscript, including the abstract.

Authors

Thank you for the positive review and the comment, we have revised the writing and the editing of the manuscript to eliminate those problems.

It is not clear why you test CO-TA, PCL-DMPA, and PCL-TA, but not CO-DMPA. Please explain it.

Authors

Our interest was to use as a base WBPU one with high biobased content (43% for CO-TA) and in particular the replacement of DMPA, which is commonly used by a molecule obtained directly form the biomass (it is a byproduct from the wine industry) was much more interesting for us.  Additionally, we have already published a study on the CO-DMPA and compared with CO-TA in a previous publication:

  1. E. Victoria Hormaiztegui, Mirta I. Aranguren and Verónica L. Mucci, “Synthesis and characterization of a waterborne polyurethane made from castor oil and tartaric acid”, European Polymer Journal, 102, (2018), pp. 151-160, https://doi.org/10.1016/j.eurpolymj.2018.03.020

Besides, the films produced from CO-DMPA were very fragile and could not be tested for mechanical properties (notice that we are not using additional chain extenders, only the triglyceride and the emulsifier and OH containing components).  Because of their fragility, it did not make sense to use this WBPU to improve the flexibility of CO-TA, so we used the ones containing PCL that contributes with a more flexible chain to the overall structure. In a previous work, we incorporate PCL together with the CO as a source of OH during the WBPU synthesis, but this strategy requires of a new synthesis for each formulation change, instead of simply mixing a fewer number of already made WBPU suspensions. [Bio-based waterborne polyurethanes reinforced with cellulose nanocrystals as coating films. M. Eugenia V. Hormaiztegui, Bernardo Daga, Mirta I. Aranguren, Verónica Mucci. Progress in Organic Coatings 144 (2020) 105649]

Please improve the quality of fig. 3 and 4 (diffraction patterns)

Authors

New versions of images have been uploaded and pasted in the manuscript.

The results and discussion section lacks discussion of the results. It is too scarce. Please improve it.

We have revised the discussion accordingly to the reviewer's suggestion and consequently new references were added to the new uploaded version of the manuscript.  We particularly thank for the comments, since we sincerely think that it has helped to improve the work.

Please carefully verify the whole manuscript for editorial and language issues, paying attention to the literature section.

Authors

The whole manuscript has been revised regarding the editorial and language issues, hopefully having corrected most of the previous errors.

Reviewer 3 Report

This work presents the fabrication of hybrid films from mixtures of castor oil and water-based polycaprolactone polyurethanes. The topic is interesting and is within the scope of the journal, however the manuscript needs some changes to have publication quality. In general, I do not recommend accepting this article in its current form. Following are my detailed suggestions for future improvements, and then the same can be accepted.

abstract graphic

1. Please build an abstract figure that explains the methodology for preparing the material.

Future perspectives.

1. The authors must create a new section before the conclusion, in this part the next researches that can enable the use of the developed material, such as its application in adsorption as an adsorbent, must be described.

Author Response

Reviewer #3

This work presents the fabrication of hybrid films from mixtures of castor oil and water-based polycaprolactone polyurethanes. The topic is interesting and is within the scope of the journal, however the manuscript needs some changes to have publication quality. In general, I do not recommend accepting this article in its current form. Following are my detailed suggestions for future improvements, and then the same can be accepted.

Authors

We thank the reviewer for the opportunity given to us to improve the manuscript

Abstract Figure:

  1. Please build an abstract figure that explains the methodology for preparing the material.

Authors

The graphical abstract was revised to include more information regarding the methodology of preparation, while maintaining (hopefully) the balance with respect to the main sections or our work.

Future Perspectives:

  1. The authors must create a new section before the conclusion, in this part the next researches that can enable the use of the developed material, such as its application in adsorption as an adsorbent, must be described.

Authors

The WBPUs were prepared with its potential application as coating for metals and/or woods, thus application as adsorbents was not considered.  However, the resistance to organic solvents (in our case, toluene was selected) is interesting and so the solubility and swelling was investigated. 

Reviewer 4 Report

1. The whole abstract needs to be rewritten. The significance and purpose of this research should be clearly presented in the abstract. The abstract must be presented in a clear way in problematic, objective, idea, description of idea, highlighting the methods, results, quantitative comparison of results with significant findings, conclusions.

2. The state-of-the-art comparisons for the proposed work are missing in this paper. Then do a critical analysis of previous research. State explicitly the shortcomings of previous research. What is positive in previous research and what is negative. Based on that, you explicitly define the goal of the research and the scientific hypothesis.

3. Highlight the novelty of your methodology.
4. The biggest shortcoming of the research is that there is no analysis of errors, analysis of sensitivity of results and analysis of uncertainty of results.
5. How did you choose the experiment setting? Elaborate experiments parameters.
7. The Conclusion section should be rewritten. Highlight your scientific contribution. Highlight the benefits of your research. Define shortcomings and future research.

Author Response

Reviewer #4

  1. The whole abstract needs to be rewritten. The significance and purpose of this research should be clearly presented in the abstract. The abstract must be presented in a clear way in problematic, objective, idea, description of idea, highlighting the methods, results, quantitative comparison of results with significant findings, conclusions.

Authors

We thank the reviewer for his careful revision of our work.  We have tried to answer to all the comments and particularly with respect to the abstract, we have rewritten most of it, introducing all the "sections" suggested by the reviewer.

  1. The state-of-the-art comparisons for the proposed work are missing in this paper. Then do a critical analysis of previous research. State explicitly the shortcomings of previous research. What is positive in previous research and what is negative. Based on that, you explicitly define the goal of the research and the scientific hypothesis.

Auhtors

We have revised this section, included new literature references and indicated what are our goals and differences with previously published works. Additionally, the specific comparisons with the several of the results of other authors have also been included or expanded with more detail in the discussion section.

  1. Highlight the novelty of your methodology.

Authors.

We have better present this point in the abstract, introduction and conclusions.

  1. The biggest shortcoming of the research is that there is no analysis of errors, analysis of sensitivity of results and analysis of uncertainty of results.

Authors

Thanks for the comments, although some of them were already done, we completed other ones.  Some of the results on the significant differences between the results are now incorporated in the main text (Table water contact angle), while others were left in the supporting information, so that the reader can have access to it, but it does not interrupt the flow of the discussion. Also, some missing error bars were included (for example in the plot of opacity, Figure 2).

We believe that the new information is very useful for the comparisons made through the text and generally confirm the trends reported, but adding a reassurance that is very much appreciated.

  1. How did you choose the experiment setting? Elaborate experiments parameters.

Authors

Since this is not our first publication on the subject, many details on the methodology of synthesis have been already published and the neat WBPU were prepared following those previous works. That is why the details of some of the syntheses are not incorporated in the present work, since it would appear as repeated from published literature. Thus, for this particular work, only the PCL-TA synthesis required to set some of the parameters of the synthesis.  Regarding the selection of temperature ranges for the studies, they are common ranges reported in the literature, as well as the selection of purge of N2 in the TGA or selected frequency of the tests in the DMA studies. 

In spite of this, we revised the Experimental section and add more information when that was possible, stressing the sentences in which the experimental setting was the result of a previous study and referring the interested reader to those particular publications.  For example, for the synthesis of other batches of similar WBPU prepared in the same or alternative reactors, the details of the synthesis and the analytic characterization of the polymers can be found in:

* M. E. Victoria Hormaiztegui, Mirta I. Aranguren and Verónica L. Mucci, “Synthesis and characterization of a waterborne polyurethane made from castor oil and tartaric acid”, European Polymer Journal, 102, (2018), pp. 151-160, https://doi.org/10.1016/j.eurpolymj.2018.03.020

The reviewer may also be interested in knowing that previously to that already published study, scouting preparations were reacted in small tubes to find the concentration of the main reactants that avoid crosslinking of the CO and TA with the IPDI.  Since the reaction is not "ideal" and secondary reactions, cyclization and reaction of the emulsifier can take place, a simple theoretical stoichiometric calculation of the quantities of the components to utilize is not enough and can give results far from those experimentally obtained. The reaction conditions used in the above previous publication resulted from that much older scouting. Similarly, this type of scouting was done to found the ratios NCO/OH and PCL/TA fitted for the newer PCL-TA synthesis.

As for examples of the parameters used in the other experimental tests, the parameters chosen were mostly extension of the one used for similar materials. For example, the CO-TA formulation has also already been used in the preparation of cellulose nanocomposites (as another form of varying the properties of the films) and in a study of the of the coating as such:

* M. E. Victoria Hormaiztegui, Verónica L. Mucci and Mirta I. Aranguren, "Composite films obtained from a waterborne biopolyurethane. Incorporation of tartaric acid and nanocellulose", 15 December 2019,  Industrial Crops and Products, 142, 111879.  https://doi.org/10.1016/j.indcrop.2019.111879.

* M. Eugenia V. Hormaiztegui, Daga Bernardo, Mirta Aranguren, Veronica Mucci, "Bio-based waterborne polyurethanes reinforced with cellulose nanocrystals as coating films", Progress in Organic Coatings. 144, 105649, 2020. https://doi.org/10.1016/j.porgcoat.2020.105649

Again and notwithstanding the above, the Experimental section was revised to add more methodology details where possible, without repeating ourselves.

  1. The Conclusion section should be rewritten. Highlight your scientific contribution. Highlight the benefits of your research. Define shortcomings and future research.

Authors.

A new section of future research was added as suggested by reviewer #3, so it was not repeated in the conclusions (except for the inclusion of a short sentence). The conclusion section was entirely revised according to the reviewer's suggestion.

Round 2

Reviewer 3 Report

the manuscript may be accepted for publication.